# Combined Toxic Effects of Lead and Glyphosate on *Apis cerana cerana*

**DOI:** 10.3390/insects15090644

**Published:** 2024-08-27

**Authors:** Yunfei Xue, Wenzheng Zhao, Qingxin Meng, Linfu Yang, Dandan Zhi, Yulong Guo, Dan Yue, Yakai Tian, Kun Dong

**Affiliations:** Yunnan Provincial Engineering and Research Center for Sustainable Utilization of Honeybee Resources, Eastern Bee Research Institute, College of Animal Science and Technology, Yunnan Agricultural University, Kunming 650201, China; xyf13183307087@163.com (Y.X.); rurosezwz@163.com (W.Z.); 13015572330@163.com (Q.M.); ylf1943349635@163.com (L.Y.); zhidand@outlook.com (D.Z.); guoyulong123@hotmail.com (Y.G.); yuedan2015@126.com (D.Y.)

**Keywords:** sublethal effects, LC50, proboscis extension reflex, detoxification enzymes

## Abstract

**Simple Summary:**

Pollinator bees are often exposed to environmental pollutants such as herbicides and heavy metals during the pollen collection process. Most of the current studies focus on the individual toxic effects of herbicides and heavy metals on bees. However, bees are generally threatened by both heavy metals and herbicides in real environments. Therefore, in this study, the common heavy metal lead and the widely used herbicide glyphosate were selected for joint toxicity analysis. We assessed the combined acute toxicities of heavy metal lead and herbicide with respect to *Apis cerana cerana* and found that they have a synergistic effect. Feeding the bees this mixture severely impaired their ability to learn and remember and thus their ability to feed. In the long term, enzyme activity and gene expression in bees were also affected. This study helps reveal the interaction of lead and glyphosate and provides a basis for the study of multiple environmental pollutants.

**Abstract:**

Glyphosate (GY) is the most widely used herbicide in agriculture worldwide. Lead is a common heavy metal in the natural environment. Honeybees, as pollinators, are exposed to these pollutants. So far, few reports have evaluated the toxic effects of GY mixed with heavy metals on honeybees (*Apis cerana cerana*). This study found that the acute toxicity of lead (LC50 = 1083 mg/L) is much greater than that of GY (LC50 = 4764 mg/L) at 96 h. The acute toxicities of the mixed substances were as follows: LC50 = 621 mg/L of lead and LC50 = 946 mg/L of GY. The combination of lead and GY was more toxic than either of the individual substances alone. Compared to the individual toxicity, combined treatment significantly affected the bees’ learning and cognitive abilities and changed the relative expression of genes related to immune defense and detoxification metabolism in *A. c. cerana*. The combination of lead and GY seriously affected the behavior and physiology of the studied honeybees. This study provides basic data for further research on the combined effects of GY and heavy metals on bee health. It also serves as a reference for effective colony protection.

## 1. Introduction

Pollinating insects are crucial for maintaining ecological balance, and one-third of global food production relies on insect pollination [1]. Honeybees, as significant pollinators, generate an economic value exceeding USD 200 billion annually through pollination [2]. Unlike *Apis mellifera*, *Apis cerana cerana* exhibits higher flight sensitivity, a longer honey collection period, enhanced adaptability, robust disease resistance, and lower feed consumption. This species thrives in harsh environments with extreme temperatures [3,4]. However, economic and societal development has led to the generation of environmental pollutants like heavy metals (lead, chromium, copper, etc.) from mining, the battery industry, and industrial parks and agricultural chemicals such as pesticides and herbicides. These contaminants pose direct and indirect risks to pollinators [2].

Glyphosate (GY), the most prevalent herbicide globally, was first marketed as Roundup [5]. Its chemical structure is C3H8NO5P, with a molecular weight of 169.1 g·mol^−1^, and it predominantly consists of isopropylamine salt (IPA salt). GY primarily inhibits the growth of plants and certain microorganisms by targeting 5-enolacetone shiki-3-phosphate synthases (EPSPS) [6]. Although studies indicate that GY is minimally toxic to animals and humans [7,8], it poses a sublethal threat to honeybees. Prolonged exposure can adversely affect bee physiology and behavior [9]. Increasing research highlights the significant effects of GY on bee gut microbiomes, development, learning, memory, and antioxidant pathways [10,11]. While bees encounter sublethal levels of GY in nature, it accumulates in colonies as bees return to their nests. Although not immediately lethal, GY may severely impact colonies over time [12].

Heavy metals are toxic substances that do not degrade in the environment and are usually present in water, air, and soil. They accumulate continuously, thus harming ecological balance [13]. Lead, a corrosion-resistant heavy metal, persists in the environment and is extensively used in the chemical, cable, battery, and radiological protection sectors. It ranks second among the twenty most harmful substances, and the World Health Organization Agency for Research on Cancer classifies it as a Group 2B carcinogen [14]. Studies have shown that lead produces reactive free radicals, damages cell structure, reduces the activity of immune system cells, inhibits normal enzyme activity, and interferes with DNA transcription and neurotransmitter release [15]. Bees encounter heavy metals mainly through air particles or by collecting nectar, pollen, or water containing these metals [16]. Recent studies have confirmed that lead increases mortality in honeybee pupae, decreases their learning and cognitive abilities, and accumulates in different body parts of these organisms [17,18].

Against this backdrop, bees often face exposure to multiple environmental pollutants during collection [19]. Jumarie et al. showed that mixtures of herbicides (GY and atrazine) and metals (cadmium and iron) affect vitamin A metabolic pathways in lab-caged bees [20]. Almasris et al. exposed winter honeybees (*A. mellifera*) to the insecticide imidacloprid, the fungicide isoproconazole, and the herbicide GY alone or in binary or ternary mixtures. They found that the different treatments differently impacted oxidative stress and metabolic and immune enzyme activities. Various concentrations of these pesticides produced varied effects [21]. Zhu et al. found that no additive or synergistic interaction occurred between clothianidin and GY in bees after mixing these two compounds [22]. Different combinations of herbicides, pesticides, and heavy metals may have different effects on bees. Thus, research into the combined effects of various chemical mixtures on bees is critically needed.

This study aims to assess the toxic effects of different treatments of lead, GY, and their combination on honeybees (*A. c. cerana*). It also analyzes the effects of these treatments on the learning and memory abilities of honeybees at sublethal concentrations. Additionally, it examines the effects on the expression of detoxification metabolism, oxidative stress, and immune-related genes as well as enzyme activity in honeybees. This study further explores the combined effect and physiological mechanisms of lead and GY toxicity with respect to honeybees. This research is intended to enrich the understanding of the toxicology of pollinators after ingesting a mixture of herbicides and heavy metals and provide important information for the study of the synergistic effects of various environmental pollutants on pollinators.

## 2. Materials and Methods

### 2.1. Honeybee Breeding

In Kunming, China (27°7′555″ N, 102°44′59″ E), four groups of sealed brood combs from *A. c. cerana* were randomly selected (one comb per group) and placed in a dark artificial climate incubator (35 ± 1 °C) with 60% RH in December 2023 to await the emergence of new honeybees. Subsequently, 0-day-old worker bees (which emerged within 24 h) were placed in nylon mesh cages measuring 10 × 10 × 10 cm^3^. Subsequently, the bee cage was placed in a dark artificial climate incubator (30 ± 1 °C 60% RH), and the worker bees were fed 50% sucrose solution. Experiments proceeded after 7 days of adaptation.

### 2.2. Chemicals and Reagents

PbCl_2_ (CAS: 7758-95-4; molecular weight, 278.11; 99.5%), GY (CAS: 1071-83.6, 95%), and caproaldehyde (CAS:66-25-1; molecular weight, 100.16; 99%) were supplied by Shanghai McLean Biochemical Technology Co., LTD. (Shanghai Chemical Industrial Park, Shanghai, China). Concentrated solutions for the lead group (PB) and glyphosate group (GY) were prepared using 50% sucrose solution and stored at 4 °C.

### 2.3. Acute Toxicity Test

Bees were collected at random from four colonies and placed in bee cages. Each concentration was replicated three times, with 20 bees per cage. Pre-experiments helped determine the concentration ranges of lead and glyphosate causing 10–90% mortality. Three treatments were designed, namely, lead (PB group, with the concentration established according to reagent purity and relative molecular weight), glyphosate (GY group, with the concentration established according to the purity of the reagent), and a combination of lead and glyphosate (PG group, high-concentration reagents from the PB and GY groups), each mixed with 50% sucrose solution. Concentration gradients for each group were set: at 3000, 1500, 1000, 750, 500, 250, and 125 mg/L for the PB group; at 8000, 4000, 2000, 1000, 500, 250, and 125 mg/L for the GY group; and at combined concentrations of 3000 + 8000, 1500 + 4000, 1000 + 2000, 750 + 1000, 500 + 500, 250 + 250, and 125 + 125 mg/L for the PG group. Control groups were offered 50% sugar water. The solutions were stored at 4 °C and replaced every 24 h. Bees were kept in an artificial climate incubator at 30 ± 1 °C and 60% RH in darkness, and mortality was assessed after 72 h and 96 h of exposure.

LC50 values for lead and GY were calculated using probit analysis. Combined toxicity was estimated using the additive index (AI) method [23]. In S = Am/A1 + Bm/B1, S is the sum of the lead and GY toxicities; A1 and B1 are the LC50s for lead and glyphosate, respectively. LC50 was determined under Am and Bm combined exposure. When S < 1, AI (the additive index) = 1/S −1.0; when S > 1, AI = S (−1) +1.0. The combined pollution effect was evaluated based on AI results: AI > 0 indicates synergy, AI < 0 indicates antagonism, and AI = 0 denotes an additive effect.

### 2.4. Chronic Toxicity Test

According to the residual amount of pollen and nectar in the natural environment being up to 56 mg/L combined with the LC50, the upper limit of the sublethal concentration of Pb was set to be about 1/10 of the LC50 (PbH = 100 mg/L) [24]. Two medium- and low-concentration groups (PbM, and PbL) were established, corresponding to concentrations of 20 and 2 mg/L, respectively. For the GY group, concentration changes (300, 150, and 75 mg/L) were selected from the first three days after GY was sprayed as the experimental concentrations and divided into GYH, GYM, and GYL groups [12]. The PG group received treatments of lead mixed with GY at concentrations of 100 + 300 mg/L, 20 + 150 mg/L, and 2 + 75 mg/L (PGH, PGM, and PGL, respectively). The control group (CK) received 50% of the normal concentration of sugar water. Each of the three treatment groups had three concentration gradients, and there was one control group. The concentrations of lead and GY in the control group were zero, and each concentration was replicated four times. Thirty bees were randomly selected from the cages after 7 d of adaptive feeding as a replicate, and four replicate experiments were conducted for each experimental concentration (from four cages, 30 bees were taken from each cage as a replicate sample). Worker bees were maintained in an artificial climate incubator (30 ± 1 °C, 60% RH) under dark conditions in the bee cage. The corresponding sugar water was replaced every 24 h, and mortality and feed intake were measured. The total exposure period lasted 14 d.

### 2.5. Proboscis Extension Reflex (PER) Experiments on Worker Bees

Field bees were collected from the bee colony and restrained to expose their entire heads based on a model developed by Tan et al. [25]. In preliminary experiments, honeybees were first starved for 2 h, and then the PER experiment was conducted using sugar water to exclude honeybees that did not present the PER. Subsequently, nine groups of sublethal concentrations were established according to the chronic toxicity test (see Section 2.4 Chronic toxicity test, PBH, PBM, PBL; GYH, GYM, GYL; PGH, PGM, and PGL) and a control group (for which both the lead and GY concentrations were 0). They were administered 10 µL of sugar water at a 50% sucrose concentration and placed in an artificial climate incubator for 1 h (30 ± 1 °C, 60% RH). The setup for the experiment involved two gases (A: volatile gas of hexaldehyde solution; B: air). A served as the conditional stimulus with sugar water feeding, while B was the control gas without a sugar water reward. A follow-up experiment was conducted 1 h after confirming that the bees had no PER reaction to airflow containing hexaldehyde. The process was as follows: bees were allowed to undergo airflow adaptation for 15 s, followed by stimulation of their antennae with hexaldehyde flow for 6 s. During this period, hexaldehyde flow was applied for 3 s; simultaneously, we rewarded the bees with sugar water for 3 s while continuing to apply the hexaldehyde flow to facilitate gas-associative learning. The reinforcement experiment was conducted every 10 min; 2 µL of hexaldehyde solution was added to the experiment setup, with a total of five reinforcement experiments performed (tests 1–5). Short-term memory ability was assessed 1 h after the last reinforcement and again 2 h later (test 7), by exposing bees to hexaldehyde gas alone for 3 s without the sugar water reward.

### 2.6. Biochemical and Molecular Tests

#### 2.6.1. Sample Preparation

Bees were collected after 14 days of exposure in the chronic toxicity test (with 9 trial groups and 1 control group; see Section 2.4 Chronic Toxicity test for further details). From each group, 30 bees were collected. Of these bees, 5 were anesthetized with ice cubes. Heads and abdomens were then collected on an anatomical dish over ice, forming one head-mixed sample and one abdomen-mixed sample, respectively. These were then placed into 1.5 mL centrifuge tubes to determine memory, detoxification, and immune-related gene expression, along with related enzyme activity. Six head samples and six abdomen samples were collected from each group and stored in a −80 °C ultra-low-temperature freezer for testing.

#### 2.6.2. Biochemical Tests

Abdominal samples were placed into a 1.5 mL centrifuge tube containing 200 μL of total protein extraction buffer RIPA lysate with 1 mM of phenylmethanesulfonyl fluoride (PMSF) [26]. Samples were ground using a hand-held tissue grinder until fully disrupted. After disruption, the supernatant was removed via centrifugation at 14,000× *g* at 4 °C for 5 min and stored at −80 °C. Total proteins isolated from tissues were quantified using the Enhanced BCA Protein Concentration Kit. For analysis, the tested sample was diluted 10 times with PBS solution. A 20 μL diluted sample and 200 μL of BCA working liquid were added to a 96-well microdrop plate and incubated at 37 °C for 30 min. Absorbance was measured at 560 nm using a thermoscientific MULTISKAN FC (Thermo Scientific, Waltham, MA, USA).

Glutathione S-transferase (GST, product number D799612-0100) and carboxylesterase (CarE, product number D799812-0100) activities were measured using commercial kits from China Shengong Biological (Shanghai) Engineering Co., LTD (Shanghai, China). As per the kits’ instructions, 5 bees served as the sample for each test. Total proteins were extracted post-centrifugation at 15,000 rpm for 10 min at 4 °C. Reagents were added as per the steps outlined, and absorbance was measured. All enzyme activities are expressed in enzyme units (U) per protein content, where one U corresponds to the degradation of one unit of substrate specified by each assay kit.

#### 2.6.3. Gene Expression Analysis

The expression of 2 memory genes (*TyrR2* and *Nmdar1*) in head samples and 6 detoxification and immunity genes (*abaecin*, *defensin*, *CYP9Q1*, *CYP9Q3*, *GSTD*, and *P450 9e2*) in abdomen samples was determined via qPCR (Table 1). The specific methods employed are as follows: Each sample was extracted with trizol RNA and converted to cDNA using a Takara reverse transcription kit. The cDNA was used as a template in the qPCR reaction. The specific reaction conditions were predenaturation at 94 °C for 30 s, followed by denaturation at 94 °C for 5 s and annealing at 60 °C for 34 s (fluorescence reading). A total of 40 cycles were completed, and data were analyzed using the 2-ΔΔCt method. Actin was used as the internal reference gene [27].

### 2.7. Statistical Analysis

Data are expressed as means ± standard error (SE). The normality of the experimental data for each group was tested using the Kolmogorov-Smirnov test, and all data were tested to show a normal distribution. Brown–Forsythe test was used to determine the homogeneity of data variance. *p* > 0.05 indicated data homogeneity. One-way analysis of variance and Tukey test (SPSS v23.0) were used to compare groups. When *p* < 0.05, the non-parametric test was used. Log-rank (Mantel-Cox) test was used to compare survival curve differences between all groups (GraphPad Prism 10).

## 3. Results

### 3.1. Toxic Effects of Lead and GY in Isolation and Combination on A. c. cerana

The data in Table 2 display the lethal toxic effects of lead and GY on honeybee workers alone and in combination. As the exposure time increased, the toxic effects intensified. After 72 h, the toxicity of lead to bees was 4.8 times greater than that of GY. At 96 h, lead was 4.4 times more toxic to bees than GY. Based on the LC50 values for lead and GY, whether used alone or in combination, the substances exhibit synergistic effects at 72 h and 96 h intervals.

### 3.2. Combined Toxic Effects of Lead and Glyphosate on Survival and Feed Intake of A. c. cerana

The mortality of the workers exposed to lead or GY alone and in combination is shown in Figure 1a. Overall, the toxicity of lead combined with glyphosate (PG) was higher than that of lead (PB group) or GY (GY group) alone. After 14 days of exposure, honeybee mortality increased with concentration, indicating a positive correlation between concentration and mortality. Significant differences were observed between the high-concentration lead group (PBH) and the high-concentration mixed group (PGH) compared with the control group. After 14 days, the cumulative feed intake of all groups (PB, GY, and PG) declined, and significant differences were noted in the PBM, PBH, GYH, and PGH groups compared with the control group. This suggests that ingesting lead and GY reduces feed intake in *A. c. cerana*.

### 3.3. Combined Toxic Effects of Lead and Glyphosate on Proboscis Extension Response of A. c. cerana

No significant difference was noted in the rate of PER instances observed among the low-concentration groups treated with PB, GY, and PG, but the rate in the low-concentration group was significantly lower than that of the control group (Figure 2a). The rate of snout elongation in the medium concentration group was also significantly lower than that in the control group. Differences in the rate of snout elongation among the medium concentration groups with memory enhancement ranged from high to low in the following order: glyphosate (GYM), lead (PBM), and lead combined with glyphosate (PGM) (Figure 2b). Neither the high-concentration lead group (PBH) nor the lead-combined-with-glyphosate group (PGH) showed significant snout elongation in conjunction with memory enhancement. As the number of experiment times increased, the GYH group showed some enhancement, but its rate of elongation was still significantly lower than that of the control group (Figure 2c).

### 3.4. Combined Toxic Effects of Lead and Glyphosate on Enzyme Activity in A. c. cerana

GST and CarE enzymes are critical detoxification enzymes in *A. c. cerana* [28]. In comparison with the control group (CK), GST enzyme activity in the three treatment groups at different concentrations was generally inhibited. GST enzyme activity in *A. c. cerana* in the high-concentration lead group (PBH) and high-concentration glyphosate group (GYH) was significantly lower than that in the CK group. Similarly, GST activity in the medium-concentration lead-combined-with-GY group (PGM) was significantly lower than that in the CK group. However, GST activity in the lead-combined-with-GY group (PGH) was not significantly different from that in the CK group but was significantly higher than that in the GYH group (Figure 3a). Regarding CarE enzyme activity, compared to the CK group, activity in the three treatment groups at varying concentrations decreased significantly. Specifically, CarE activity in the medium-concentration group changed notably, with activity in the medium-concentration lead-combined-with-GY group (PGM) being significantly lower than that in the medium-concentration glyphosate group (GYM) (Figure 3b). Overall, GST and CarE enzyme activities in *A. c. cerana* were reduced in most cases when the corresponding bees were exposed to various concentrations of lead and glyphosate. However, GST activities were higher when lead and GY were combined at high concentrations than when administered separately.

### 3.5. Combined Toxic Effects of Lead and Glyphosate on Gene Expression in A. c. cerana

#### 3.5.1. Expression of Memory Genes

The *Nmdar1* and *TyrR2* genes are closely related to learning and memory in insects [29]. Compared with the control group, *Nmdar1* expression increased significantly in the low (PBL)- and medium (PBM)-lead-concentration groups. It decreased significantly in the medium-glyphosate-concentration (GYM) and medium-lead–glyphosate-combination-concentration (PGM) groups. No significant difference in *Nmdar1* expression was observed between the other groups and the control group. Furthermore, compared to the low-, medium-, and high-lead concentration groups (PBL, PBM, and PBH), *Nmdar1* expression significantly decreased in all the lead–glyphosate combination groups (PBL, PGM, and PGH). Its expression was also significantly lower in the lead–glyphosate low-concentration group (PGL) compared to the glyphosate low-concentration group (GYL) (Figure 4a). No significant difference in *TyrR2* expression existed between any feeding groups and the control group. In the lead–glyphosate combination group, no significant difference in *TyrR2* expression was found in the low- and medium-concentration groups (PBL, PGM) compared with the low- and medium-concentration lead groups (PBL, PBM). However, expression significantly increased in the high-concentration lead–glyphosate group (PGH) compared to the high-concentration lead group (PBH). Lead combined with glyphosate in the low-, medium-, and high-concentration groups (PBL, PGM, PGH) showed no significant differences compared to the corresponding glyphosate groups (GYL, GYM, GYH).

#### 3.5.2. Expression of Abdominal Immunity and Detoxification Genes

*Abaecin* and *Apidaecin* are the main antimicrobial peptides in *A. c. cerana*, protecting bees from foreign pathogenic substances [30]. *Abaecin* expression was significantly enhanced in the high-concentration lead group (PBH) and the medium-concentration mixed lead group (PGM). Expression in the bees exposed to lead combined with glyphosate (PGM) differed significantly from that of the bees exposed to lead (PBM) and glyphosate (GYM) alone in the medium-concentration group. In contrast, the bees in the group exposed to lead combined with glyphosate (PGH) differed only from those exposed to glyphosate (GYH) in the high-concentration group (Figure 5a). Compared with the control group, *Apidaecin* expression was significantly decreased in the medium-concentration lead (PBM), medium-concentration mixed lead (PGM), and high-concentration mixed lead (PGH) groups. Additionally, expression in the low-concentration mixed group (PGL) was significantly up-regulated compared with that of the low-concentration lead group (PBL), while expression in the high-concentration mixed group (PGH) was significantly down-regulated compared with that of the high-concentration lead group (PBH) (Figure 5b).

Regarding the *GSTD1* gene, gene expression levels in the PB, GY, and PG groups increased with concentration. The expression levels in the medium-concentration mixed group (PGM) and the medium-concentration glyphosate group (GYM), as well as in the high-concentration mixed group (PGH) and the high-concentration glyphosate group (GYH), were significantly higher (Figure 5c). Compared with the control group, *Cyp9Q1* gene expression was significantly lower in the low-concentration glyphosate group (GYL) and the mixed medium- and high-concentration groups (PGM, PGH). Additionally, expression in the mixed group (PGM) at a medium concentration was significantly lower compared with that of the groups exposed to either lead or glyphosate (PBM, GYM) alone (Figure 5d). The expression of *Cyp9Q3* in the low- and high-concentration lead groups (PBL, PBH) and medium- and high-concentration glyphosate groups (GYM, GYH) was significantly higher compared with that of the control group (CK). The expression levels in the low-concentration mixed group (PGL), low-concentration lead group (PBL), and high-concentration mixed group (PGH) were significantly lower compared with those of the groups exposed to high concentrations of lead or glyphosate alone (PBH, GYH) (Figure 5e). For the *P450 9e2* gene, its expression in the low-concentration lead group (PBL) (Figure 5f) and medium- and high-concentration glyphosate groups (GYM, GYH) was significantly higher compared with that in the control group (CK), and the expression level in the low-concentration lead group (PBL) and medium- and high-concentration glyphosate groups (GYM, GYH) was significantly higher compared with that in the mixed group.

## 4. Discussion

*Apis cerana cerana* often faces exposure to various environmental pollutants simultaneously, posing serious threats to its survival. GY and lead may accumulate in individual bees and colonies, producing lethal effects. However, the interaction between these pollutants remains uncertain. We evaluated the combined effects of lead and GY in an acute toxicity study on *A. c. cerana* for the first time. In previous studies, it was determined that the lethal concentration of glyphosate (GY) for western honeybees was 4–20 mg/L [31], and the LC50 of lead (PB) for western honeybees was 345 mg/L. In the acute toxicity test involving *A. c. cerana*, the LC50 for 48 h ranged from 6 to 8 g/L. We determined that the LC50 for 96 h with a median lethal concentration of lead (PB) was 1083 mg/L. The 96 h LC50 for GY was 4764 mg/L, significantly different from that of western bees, potentially due to the resistance of *A. c. cerana* [32]. The LC50 of lead in the combined lead–glyphosate group (PG) at 96 h was 621 mg/L, which is nearly half that of that of lead alone, i.e., 1083 mg/L. Additionally, the LC50 of GLY decreased to 946 mg/L from 4764 mg/L of GY alone, indicating a greater threat posed by lead combined with GY than GY alone. In the chronic toxicity test, the mortality in the high-concentration lead (PBH) and the high-concentration mixed group (PGH) significantly differed from that of the control group. This, combined with the reduced LC50 concentration of GY, suggests an increased toxic effect of lead and GY on bees. Our results are consistent with previous reports showing that exposure to lead and GY alone alters bees’ feeding behavior. Feed intake was higher in the mixed group (PG) than in the PB group, potentially because PB reduces the palatability of sugar water to bees. However, a PB and GLY mixture enhances adsorption and palatability [18,33]. Analysis of acute and chronic toxicity in *A. c. cerana* revealed that lead’s toxicity is much greater than glyphosate’s toxicity. The mixture of these two produces a synergistic toxic effect, as calculated via the AI index. Given the multiple threats pesticides and heavy metals pose to pollinators, it is crucial to explore the combined threat to pollinators.

When honeybees are exposed to pesticides or heavy metals, a low PER value indicates reduced learning and cognitive abilities. After enhancing the memory abilities of the bees, we conducted a PER experiment to assess if lead combined\ with-GY impacts the memory of honeybees. The experiment conducted in December showed a reduced PER rate across all groups, without altering the experimental outcomes. Low and medium concentrations of lead (PBL, PBM) led to a notable decline compared to the control group (CK), while high concentrations (PBH) elicited minimal responses, suggesting a decrease in learning and cognition with the increasing lead levels. In the GY group, five reinforcement sessions indicated retained learning capacity and similar PER responses. However, with rising concentrations in subsequent tests, memory retention deteriorated. Previous research indicates that GY impairs olfactory memory in bees without influencing cognition, and our findings confirm learning remains stable under GY exposure, though memory declines as the concentration rises [34]. In the mixed group (PG), learning and memory at medium and high concentrations (PGM, PGH) significantly deteriorated compared to the groups with single-toxicant exposure. The low-concentration group (PGL) showed comparable results to the groups with single exposures, possibly due to lead–Gy complex formation [33]. Short-term exposure to sublethal doses of lead and GY does not affect learning, but prolonged exposure might reduce feeding behaviors.

When bees are affected by external pollution, it is important to understand their detoxification mechanisms [35,36]. For example, glutathione transferase (GST) and carboxylesterase (CarE), both detoxifying enzymes in insects, are involved in bee detoxification. CarE is the primary enzyme in the first stage. GST plays a major role in the second stage [37,38]. In this study, GST enzyme activity was significantly reduced in the PBH, GYH, and PGH groups, lowering the detoxification capacity of *A. c. cerana*. Han et al. observed a significant increase in GST activity in bee foragers after short-term (1d) exposure to acetamidine and procycloconazole. However, long-term exposure led to GST inhibition [39]. Compared to the control group, CarE activity also decreased significantly, likely due to long-term exposure.

Long-term exposure to pesticides affects the learning and cognitive ability of bees. *Nmdars* comprise two subunits, *Nmdar1* and *Nmdar2*, with *Nmdar1* widely expressed in the brains of bees as the predominant subunit [29]. Among *Nmdar1* genes, only the PB group exhibited high expression. In contrast, the GY group and PG group showed reduced expression. According to Hernandez et al., GY impacts the memory but not the cognitive ability of bees [34]. Li et al. revealed that the heavy metal cadmium reduced Nmdar1 expression in fruit flies and bees [40]. *Abaecin* and *Apidaecin*, two antimicrobial peptides in bees, are crucial for humoral immunity [30]. As exposure to lead and GY increased, a decrease in gene expression was observed, potentially leading to reduced immune capacity in bees. Li and Motta et al. noted that both the heavy metal Cd and herbicides harmed the immune capacity of bees [40,41]. The significant increase in gene expression observed in a few groups could be due to bee hypersensitivity [42]. The GST gene is essential for detoxification metabolism and oxidative stress, while CYP family genes aid in pesticide detoxification, and P450s are multifunctional enzymes crucial in the detoxification or activation of insecticides [43,44,45]. As expected, exposure to lead and GY in *A. c. cerana* triggered an increase in detoxification gene expression. This rise in detoxification gene expression indicates that bees initiate a defense response against lead and GY.

## 5. Conclusions

The toxicity of lead to *A. c. cerana* was higher than that of GY, and the two had an acute synergistic effect. Our results indicate that long-term exposure to lead and GY at sublethal concentrations reduces the survival and feed intake of honeybees and severely impairs their learning and memory, thus altering feeding behavior. The expression levels of *Abaecin*, *Apidaecin*, *GSTD1*, *Cyp9Q1*, *Cyp9Q3*, and *P450 9e2*, related to immune response and detoxification metabolism, were significantly affected in the mixed group compared to those in the single group. In summary, these findings contribute to understanding the combined toxicity of lead and GY to honeybees and offer a reference for understanding the mixed toxicity of various environmental pollutants.

## Figures and Tables

**Figure 1 insects-15-00644-f001:**
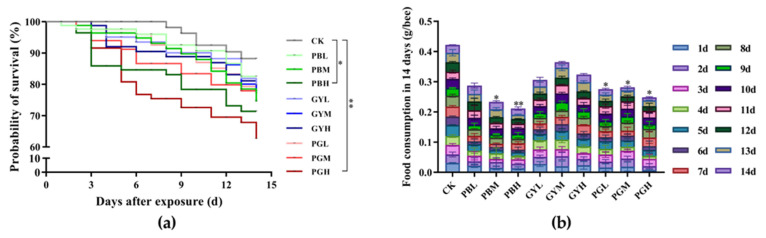
Survival rate and feed intake of *A. c. cerana* at 14 days after exposure to sublethal concentrations of lead, glyphosate, and their combination. (**a**) Survival rate changes in PB, GY, and PG groups after 14d exposure. (**b**) Cumulative feed intake of PB, GY, and PG groups after 14 days of exposure. The results are expressed as means ± SE. Compare each treatment with the CK. * *p* < 0.05; ** *p* < 0.01. PB = Pbcl2; GY = glyphosate; PG = lead combined with glyphosate; CK = 50% sucrose sugar water. L, M, and H are low, medium, and high doses of PB, GY, and PG, respectively.

**Figure 2 insects-15-00644-f002:**
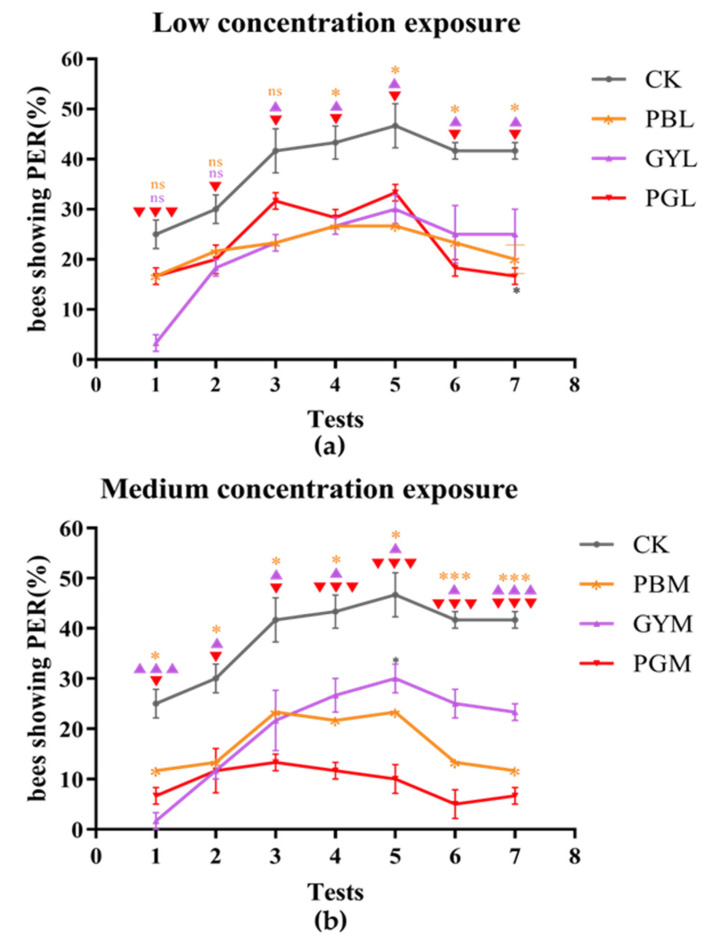
Effects of lead, glyphosate, and their combination on proboscis extension response of *A. c. cerana* compared with control group. After 8 tests, the first 5 tests were intensive memory experiments conducted at intervals of 10 min, and the 6th and 7th tests were short-term memory experiments conducted at intervals of 1 h. (**a**) Under acute exposure, the PER of the low-dose PB, GY, PG, and control groups. (**b**) Under acute exposure, the PER of the medium-dose PB, GY, PG, and control groups. (**c**) Under acute exposure, the PER of the high-dose PB, GY, PG, and control groups. * *p* < 0.05; ** *p* < 0.01; *** *p* < 0.01. PB = Pbcl2; GY = glyphosate; PG = lead combined with glyphosate; CK = 50% sucrose sugar water. L, M, and H are low, medium, and high doses of PB, GY, and PG, respectively.

**Figure 3 insects-15-00644-f003:**
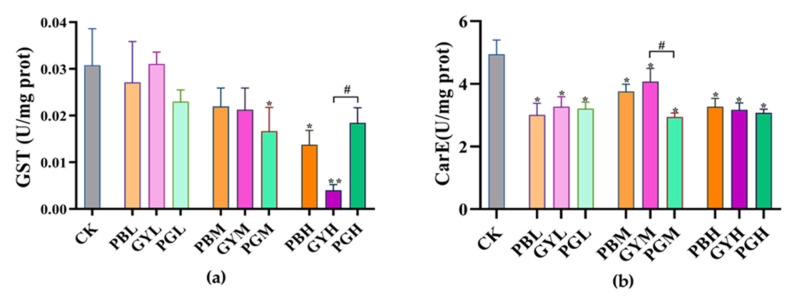
Toxic effects of lead, glyphosate, and their combination on abdominal enzyme activity of *A. c. cerana*. The results are given as means ± SE. (**a**): The expression of GST enzyme activity. (**b**) The expression of CarE enzyme activity. * indicates a significant difference between each treatment group and the control group; # indicates a significant difference between the lead-combined-with-glyphosate group and the groups exposed to either substance alone. */# *p* < 0.05, ** *p* < 0.01. PB = Pbcl2; GY = glyphosate; PG = lead combined with glyphosate; CK = 50% sucrose sugar water. L, M, and H are low, medium, and high doses of PB, GY, and PG, respectively.

**Figure 4 insects-15-00644-f004:**
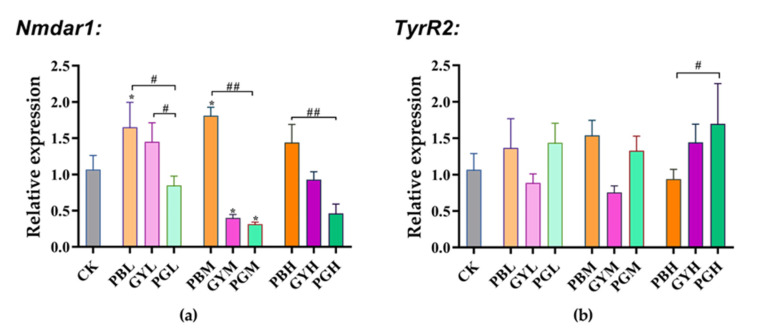
Toxic effects of lead, glyphosate, and their combination on the expression of learning and memory genes in *A. c. cerana*. The results are given as means ± SE. (**a**): Gene expression of *Nmdar1*. (**b**): Gene expression of *TyrR2.* * indicates a significant difference between the drug group and the control group; # indicates a significant difference between the lead-combined-with-glyphosate group and the groups exposed to either substance alone. */# *p* < 0.05, ## *p* < 0.01. PB = Pbcl2; GY = glyphosate; PG = lead combined with glyphosate; CK = 50% sucrose sugar water. L, M, and H are low, medium, and high doses of PB, GY, and PG, respectively.

**Figure 5 insects-15-00644-f005:**
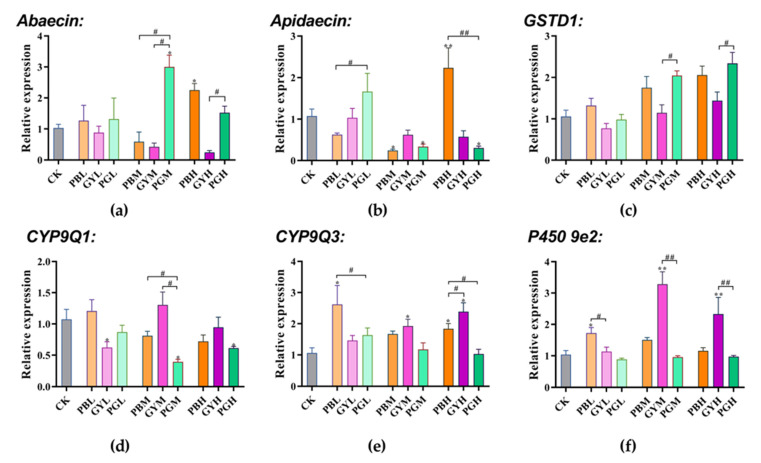
Toxic effects of lead, glyphosate, and their combination on genes related to immunity and detoxification in *A. c. cerana*. The results are given as means ± SE. * *p* < 0.05 indicates that there was a significant difference compared with the control group; # *p* < 0.05 indicates a significant difference between the lead-combined-with-glyphosate group and the groups exposed to either substance alone. */# *p* < 0.05, **/## *p* < 0.01. PB = Pbcl2; GY = glyphosate; PG = lead combined with glyphosate; CK = 50% sucrose sugar water. L, M, and H are low, medium, and high doses of PB, GY, and PG, respectively.

**Table 1 insects-15-00644-t001:** The sequences of primers.

Gene	Primer Sequence (5′ to 3′)
*Nmdar1*	F:CGGGCTTCGATAGCTGCTCAAC R:CTCGACCAGGACTTCGTGGAGT
*TyrR2*	F:CGACCACGACAACGGTTTACCA R:CACGCCCATTATCACGCCCAAT
*Abaecin*	F:CGCAGCGTTCGCATATGTACCA R:ATGGTCCCTGACCAGGAAACGT
*Apidaecin*	F:CCTTTATAGTCGCGGTATTTGG R:AGGTTCAGCTTCCGGTTCAG
*CYP9Q1*	F:GGCTTCAGGAGGAGATCGACGA R:ACCTCATCGCCTCGCTCATCA
*CYP9Q3*	F:ACTCCATCACCTCCGTCACCAC R:CACACCGTCGTTGCTCCTCAA
*P450 9e2*	F:ATGGACGGCGAGCAATGGAAG R:TGGACAGGTGATGGGCGAATCT
*GSTD1*	F: CATTGACGGCTGCTGCTCT R: TTCAGGCTTCAACTGTTCCCTAT
*Arp(actin)*	F:GGATTCGGGTGACGGTGTTTCG R:GCCAGCCAAGTCCAAACGAAGA

**Table 2 insects-15-00644-t002:** Individual and combined toxicity of lead and glyphosate with respect to *A. c. cerana* workers.

Exposure Time (h)	Glyphosate Alone			Lead Chloride Alone			Glyphosate Combined			Lead and Chloride Combined			Additive Index (AI)
	df	*p*	LC50 (95% FL) mg a.i. L^−1^	df	*p*	LC50 (95% FL) mg a.i. L^−1^	df	*p*	LC50 (95% FL) mg a.i. L^−1^	df	*p*	LC50 (95% FL) mg a.i. L^−1^	
72	6	0.36	7121 (6089–8657)	6	0.34	1483 (1341–1659)	6	0.18	1448 (1263–1672)	6	0.39	769 (701–847)	0.43
96	6	0.23	4764 (4184–5505)	6	0.31	1083 (985–1203)	6	0.48	946 (830–1083)	6	0.18	621 (566–682)	0.30

Notes: df stands for degrees of freedom; *p* > 0.05 indicates that the model has a high degree of fitting. Other content is specified in Appendix A.

## Data Availability

The raw data supporting the conclusions of this article will be made available by the authors on request.

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
