# Peer review of "Combined Toxic Effects of Lead and Glyphosate on Apis cerana cerana"

_insects, 2024, doi:10.3390/insects15090644_

Round 1
Reviewer 1 Report
Comments and Suggestions for Authors
Xue et al. report the results of a laboratory study on the toxicity of lead and glyphosate alone and in mixtures on the Asian honeybee.
MAIN POINTS
I. The main issues that I had with this manuscript revolve around the readability of the figures. I was unable to read most of the Figures which made reviewing the manuscript difficult.
II. The authors need to present evidence that the experimental concentrations of lead and herbicide are likely to be encountered by bees in agricultural settings.
III. It was also unclear to me whether the experimental concentrations and LC50 values provided in mg/L referred to the elemental lead component and pure glyphosate or the experimental preparations (PbCl2, and possibly a salt of glyphosate?).
IV. The authors need to ascertain that their data were suitable for ANOVA in several places.
V. I wrote numbered points and suggestions on a scanned copy of the manuscript.
Numbered points (see scanned file)
1. I wonder if the title could be improved by adding the words "Acute and chronic toxic effects of...." instead of "combined".
2. You mean ....most current studies focus on the toxic effects of herbicides and heavy metals individually on bees. Reword.
3. Reword. The combination of lead and GY was more toxic than either of the individual substances alone.
4. What does "mixed toxicity of genes" mean? Reword.
5. Do not use keywords that are already in the title.
6. Pb group concentrations. Are these the concentrations of LEAD or LEAD CHLORIDE? The Mr of PbCl is 278 g/mol of which 207 g is lead and the rest is chlorine. So do the indicated concentrations refer to the lead element or the compound?
The same applies to glyphosate which is usually sold is a the isopropylamine salt.
7. What was the justification for setting the upper limit of lead (PbCl2?) to 100 mg/L – please provide references to this an a normal environmental concertation in agricultural settings.
8. What is the normal agricultural environmental concentration range of GY? Please provide references.
9a, I have never heard of hexal gas or hexal solution. Please clarify.
9b. Please provide full names of the genes at first use.
10. How did you ensure that the data met the equality of variance necessary for ANOVA?
The same goes for normality in the data.
11. The text in Table 2 is tiny and very difficult to read.
12. Please provide the Chi2 values for the goodness-of-fit tests done on Probit regressions. These can be presented as supplemental Tables together with the slopes of the regressions ± SE please.
13. I did not understand this text and did not see an asterisk in Table 2.
14. I found Figure 1b too small to read.
15. I could not see the asterisks clearly in Figure 1b. Please increase size and use a higher resolution figure. I think MDPI Insects states that all figures should be 600 dpi.
16. I was unable to read Fig 2. Please increase size and use 600 dpi.
17. I was unable to read Fig 3. Please increase size and use 600 dpi.
18. I was unable to read Fig 4. Please increase size and use 600 dpi. Also please state what the y-axis indicates.
19. I was unable to read Fig 5. Please increase size and use 600 dpi. Also please state what the y-axis indicates.
20. This paragraph simply restates information from the Introduction. Please delete and focus on the interpretation of the main findings.
21. You cannot justify reporting LC50 values to a decimal place as you used very wide range of concentrations and few replicates.
22. I did not understand this sentence, please reword.
23. Please describe the role of lead-glyphosate complexes and how this could have affected the results. Does this not happen at higher concentrations? Why not?
24. You are saying that CCD is due to lead and Glyphosate??? You have no evidence for this.
25. I did not understand this sentence, please reword.
26. Please describe the situation in which foraging bees would be exposed to the experimental concentrations of lead and glyphosate in agricultural settings and provide references in support. This is of mayor importance to the validity of your findings.
27. The references need to be formatted according to journal guidelines.

Minor editing.
Author Response
Response to Decision Letter and Reviewer Comments
Dear Editors and Referees,
We would like to express our sincere appreciations to you for giving us the opportunity to revise the manuscript and to the referees for their constructive comments concerning our manuscript (manuscript ID: insects-3167197). Based on the referees' comments, we have revised the manuscript, and the parts modified are marked in red. Here are our responses to reviewer comments:
Reviewer #1
MAIN POINTS
Question #1: The main issues that I had with this manuscript revolve around the readability of the figures. I was unable to read most of the Figures which made reviewing the manuscript difficult.
Response: Thank you for the constructive comments and suggestions. We have modified the image to make it more readable.
Question #2: The authors need to present evidence that the experimental concentrations of lead and herbicide are likely to be encountered by bees in agricultural settings.
Response: We appreciate your insightful comment. We completed the introduction section to add concentrations of lead and glyphosate in experiments that may be encountered in the environment. Specific references are as follows:
- Xun, E.; Zhang, Y.; Zhao, J.Guo, J. Heavy metals in nectar modify behaviors of pollinators and nectar robbers: Consequences for plant fitness. Environmental Pollution2018,242, 1166-1175.
- Odemer, R.; Alkassab, A.T.; Bischoff, G.; Frommberger, M.; Wernecke, A.; Wirtz, I.P.; Pistorius, J.Odemer, F. Chronic high glyphosate exposure delays individual worker bee (apis mellifera l.) development under field conditions. Insects2020,11, 664.
Question #3: It was also unclear to me whether the experimental concentrations and LC50 values provided in mg/L referred to the elemental lead component and pure glyphosate or the experimental preparations (PbCl2, and possibly a salt of glyphosate?).You mean ....most current studies focus on the toxic effects of herbicides and heavy metals individually on bees. Reword.
Response: Thank you for your constructive feedback. In the manuscript, we restated the specific components of the experimental concentration and LC50 value provided in mg/L and revised the questions raised, please see lines 133-137.
Question #4: The authors need to ascertain that their data were suitable for ANOVA in several places.
Response: Thank you for your observation. We have completed the addition of ANOVA, please see lines 213.
Question #5: I wrote numbered points and suggestions on a scanned copy of the manuscript.
Response: Thank you for pointing it out. We have addressed these issues on the basis of the points provided
Numbered points (see scanned file)
Question #1: I wonder if the title could be improved by adding the words "Acute and chronic toxic effects of...." instead of "combined".
Response: We appreciate your insightful comment. This study focuses on acute toxicity and chronic toxicity to bees by means of a more comprehensive description of combined toxicity. This paper focuses more on the comparison of combined toxicity and individual toxicity.
Question #2: You mean ....most current studies focus on the toxic effects of herbicides and heavy metals individually on bees. Reword.
Response: Thank you for the constructive comments and suggestions. We have redescribed it in the manuscript,please see lines 15.
Question #3: Reword. The combination of lead and GY was more toxic than either of the individual substances alone.
The combination of lead and GY seriously affects the behavior and physiology of honeybees.
Response: We appreciate your insightful comment. According to your suggestion, we have revised the contents of lines 30
Question #4: What does "mixed toxicity of genes" mean? Reword.
Response: Thank you for the constructive comments and suggestions. What I want to express is the effect of lead and glyphosate mixture on gene expression, which has been reexpressed in the original manuscript.
Question #5: Do not use keywords that are already in the title.
Response: We appreciate your insightful comment. We've refocused our keywords.
Question #6: Pb group concentrations. Are these the concentrations of LEAD or LEAD CHLORIDE? The Mr of PbCl is 278 g/mol of which 207 g is lead and the rest is chlorine. So do the indicated concentrations refer to the lead element or the compound?
The same applies to glyphosate which is usually sold is a the isopropylamine salt.
Response: Thank you for your constructive feedback. This is the concentration of lead ions, We used the relative molecular weight of PbCl and the purity of the reagent to obtain the corresponding concentration of the specific content configuration of lead. Glyphosate is a commercial glyphosate we configure according to the corresponding purity.
Question #7: What was the justification for setting the upper limit of lead (PbCl2?) to 100 mg/L – please provide references to this an a normal environmental concertation in agricultural settings.
Response: Thank you for your valuable feedback.In the specific reference, the concentration of lead reached 56mg/L, and the reason for setting 100mg/L was to use 1/10 of LC50 measured by us to complete the setting.Specific references are as follows:
- Xun, E.; Zhang, Y.; Zhao, J.Guo, J. Heavy metals in nectar modify behaviors of pollinators and nectar robbers: Consequences for plant fitness. Environmental Pollution2018,242, 1166-1175.
Question #8: What is the normal agricultural environmental concentration range of GY? Please provide references.
Response: Thank you for your observation. In the reference, the concentration changes of 300, 150, and 75mg/L in the first three days after spraying commercial glyphosate were taken as the experimental concentration.
Question #9a: I have never heard of hexal gas or hexal solution. Please clarify.
Response: We appreciate your insightful comment. According to your suggestion, we have reformulated "hexal" to "hexaldehyde"
".
Question #9b: Please provide full names of the genes at first use.
Response: According to your suggestion, we have revised the contents of lines 202.
Question #10: How did you ensure that the data met the equality of variance necessary for ANOVA?
The same goes for normality in the data.
Response: We sincerely thank you for pointing this out. Our response to your question has been revised in Material Method 2.6
Question #11: The text in Table 2 is tiny and very difficult to read.
Response: Thank you for your insightful comment. We have resized Table 2 for ease of reading.
Question #12: Please provide the Chi2 values for the goodness-of-fit tests done on Probit regressions. These can be presented as supplemental Tables together with the slopes of the regressions ± SE please.
Response: We appreciate your insightful comment. We have modified the annotations in Table 2 and added the content you proposed in supplementary Table S2
Question #13: I did not understand this text and did not see an asterisk in Table 2.
Response: Thank you for your constructive feedback. We have completed the modification of the annotation problem.
Question #14: I found Figure 1b too small to read.
Response: We appreciate your insightful comment. We have updated the image to 600dpi and completed the adjustment of the content to increase.
Question #15: I could not see the asterisks clearly in Figure 1b. Please increase size and use a higher resolution figure. I think MDPI Insects states that all figures should be 600 dpi.
Response: We appreciate your insightful comment. We finished updating the figures to 600dpi.
Question #16: I was unable to read Fig 2. Please increase size and use 600 dpi.
Response: We appreciate your insightful comment. We finished updating the figures to 600dpi.
Question #17: I was unable to read Fig 3. Please increase size and use 600 dpi.
Response: We appreciate your insightful comment. We finished updating the figures to 600dpi.
Question #18: I was unable to read Fig 4. Please increase size and use 600 dpi. Also please state what the y-axis indicates.
Response: We appreciate your insightful comment. We finished updating the graph to 600dpi and added the y-axis indicates.
Question #19: I was unable to read Fig 5. Please increase size and use 600 dpi. Also please state what the y-axis indicates.
Response: We appreciate your insightful comment. We finished updating the graph to 600dpi and added the y-axis indicates.
Question #20: This paragraph simply restates information from the Introduction. Please delete and focus on the interpretation of the main findings.
Response: Thank you for your constructive feedback. We have refined this paragraph.
Question #21: You cannot justify reporting LC50 values to a decimal place as you used very wide range of concentrations and few replicates.
Response: We sincerely thank you for pointing this out. We have finished removing the decimal point of LC50.
Question #22: I did not understand this sentence, please reword.
Response: Thank you for your constructive feedback. We have rephrased the sentence.
Question #23: Please describe the role of lead-glyphosate complexes and how this could have affected the results. Does this not happen at higher concentrations? Why not?
Response: Thank you for your constructive feedback. We hypothesized that the increased concentration of lead would affect the formation of complexes.
Question #24: You are saying that CCD is due to lead and Glyphosate??? You have no evidence for this.
Response: Thank you for your constructive feedback. According to your suggestion, we have revised the contents.
Question #25: I did not understand this sentence, please reword.
Response: Thank you for your insightful comment. According to your suggestion, we have revised the contents of lines 423 in Discussion.
Question #26: Please describe the situation in which foraging bees would be exposed to the experimental concentrations of lead and glyphosate in agricultural settings and provide references in support. This is of mayor importance to the validity of your findings.
Response: Thank you for your constructive feedback. Lead concentration in pollen or nectar in natural environment can reach up to 56mg/L. In the first three days of use of the corresponding commercial glyphosate, the concentration changes were 300mg/L, 150mg/L, and 75mg/L. The references are as follows
- Xun, E.; Zhang, Y.; Zhao, J.Guo, J. Heavy metals in nectar modify behaviors of pollinators and nectar robbers: Consequences for plant fitness. Environmental Pollution2018,242, 1166-1175.
- Odemer, R.; Alkassab, A.T.; Bischoff, G.; Frommberger, M.; Wernecke, A.; Wirtz, I.P.; Pistorius, J.Odemer, F. Chronic high glyphosate exposure delays individual worker bee (apis mellifera l.) development under field conditions. Insects2020,11, 664.
Question #27: The references need to be formatted according to journal guidelines.
Response: We appreciate your insightful comment. We formatted the references according to the journal guidelines.
Thank you very much for your attention and time. We are looking forward to hearing from you.
Yours sincerely,
Yunfei Xue
August 21, 2024
Yunnan Agricultural University, Kunming 650201, China.
Phone: +86 13183307087
Email: xyf13183307087@163.com

Reviewer 2 Report
Comments and Suggestions for Authors
The authors investigated combined toxic effects of two common environmental pollutants, lead and glyphosate (a widely used herbicide) on honeybee’s physiology and behavior. Synergistic negative effects of various combinations of adverse environmental factors have been studied in numerous insects. Therefore the basic novelty and the potential theoretical value of the present study are rather low. However, for the practice, each particular case undoubtedly deserves to be investigated. Thus, the results of this study can be of practical importance for beekeepers and for the specialists of environment protection and hence the manuscript can be published. However, some flaws should be fixed and mistakes should be corrected before publication (see my comments below). In addition, the authors should carefully check the whole text for misprints and errors.
Line 17: Quotes signs around "lead" and "glyphosate" are not needed here.
Line 75: Apis mellifera (Latin name of an insect) should be in Italics font.
Line 84, 97, 215, 222, 235, etc. Apis cerana cerana (Latin name of an insect) should be in Italics font. Besides, the generic name “Apis” can be often abbreviated as “A.”
Line 96 and 97: Sorry, it is not quite clear to me what you mean “mature lidded spleen”. I failed to find this combination of words in any other scientific paper. Most probably, it will be also new and not clear for most potential readers. The term “spleen” is commonly used for an abdominal organ involved in the production and removal of blood cells in vertebrates. Possibly, you mean "mature combs"? Comb is a structure of hexagonal cells of wax, made by bees to store honey and eggs. Please, either correct or explain what you mean “mature lidded spleen” in details.
Line 101: Please, indicate also photoperiod (L:D) or darkness (as in the subsequent regimes).
Line 153: Sorry, another not commonly used term "no-snout phenomenon". Please, kindly explain it to less educated readers. It also necessary to provide references to earlier publications concerned this phenomenon.
Line 207: Please, include (Scharlaken et al., 2008) in the list of references (I can’t find this author in your list) and use the reference number.
Line 216: These data are not in Table 1 but in Table 2. Please, correct the table number.
Line 217: You wrote “significantly” but I do not see in Table 2 any data on the significance of the difference between 72 and 96 h exposures (such as p < .....). If you mean that 95% confidence intervals are not overlapped, please, explain it clearly.
Table 2: Please, indicate in the head of the table, the results of what test are given and explain all abbreviations. All “p” in the table are >0.05. Does it mean the absence of any significant difference between the compared groups?
Line 223: Footnote is labeled with the asterisk, but I see no asterisks in Table 2.
Figure 1a: In the legends (line 239) it is stated that the results are mean ±SE but I don’t see any SE in this graph.
Figure 1b: I guess that the asterisks mean the difference from the control group CK but this should be clearly indicated in the legends.
Figure 2: Possibly because of very low quality of these pictures, I can’t see any asterisks. Anyway, in the legends (line 262) it should be indicated that the difference between control and experimental groups are given (as it is indicated in the legends to Fig. 3, lines 284-285).
Line 283: Not “FFigure” but “Figure”.
Lines 302-303: Do not start a new paragraph here.
Lines 446-447: Delete the end of this paragraph that is the fragment of the sentence from the journal template description.
Author Response
Response to Decision Letter and Reviewer Comments
Dear Editors and Referees,
We would like to express our sincere appreciations to you for giving us the opportunity to revise the manuscript and to the referees for their constructive comments concerning our manuscript (manuscript ID: insects-3167197). Based on the referees' comments, we have revised the manuscript, and the parts modified are marked in red. Here are our responses to reviewer comments:
Reviewer #2
Question #1: Line Quotes signs around "lead" and "glyphosate" are not needed here.
Response: We gratefully appreciate your careful checks and reminders and have revised this.
Question #2: Line Apis mellifera (Latin name of an insect) should be in Italics font.
Response: We appreciate your careful inspection and reminder. According to the recommendations, the revised manuscript has been thoroughly rechecked and some errors have been corrected
Question #3: Line84, 97, 215, 222, 235, etc. Apis cerana cerana (Latin name of an insect) should be in Italics font. Besides, the generic name “Apis” can be often abbreviated as “A.”
Response: We feel great thanks for your professional review work on our article. The "apis" for the latter part has been shortened to "a".
Question #4: Line 96 and 97: Sorry, it is not quite clear to me what you mean “mature lidded spleen”. I failed to find this combination of words in any other scientific paper. Most probably, it will be also new and not clear for most potential readers. The term “spleen” is commonly used for an abdominal organ involved in the production and removal of blood cells in vertebrates. Possibly, you mean "mature combs"? Comb is a structure of hexagonal cells of wax, made by bees to store honey and eggs. Please, either correct or explain what you mean “mature lidded spleen” in details.
Response: Thank you for the constructive comments and suggestions. We did not explain it clearly and have corrected it to read sealed brood combs in the manuscript.
Question #5: Line 101: Please, indicate also photoperiod (L:D) or darkness (as in the subsequent regimes).
Response: We appreciate your insightful comment. In our experiment, bees were kept in a dark incubator throughout the whole process, which was also modified in the resubmitted manuscript.
Question #6: Line 153: Sorry, another not commonly used term "no-snout phenomenon". Please, kindly explain it to less educated readers. It also necessary to provide references to earlier publications concerned this phenomenon.
Response: Thank you for your constructive feedback. This content has been revised “without PER” in the manuscripts
Question #7: Line 207: Please, include (Scharlaken et al., 2008) in the list of references (I can’t find this author in your list) and use the reference number.
Response: Thank you for your valuable feedback. We have finished adding to the manuscript.
Question #8: Line 216: These data are not in Table 1 but in Table 2. Please, correct the table number.
Response: Thank you for your observation. We have completed the revision in the manuscript.
Question #9: Line 217: You wrote “significantly” but I do not see in Table 2 any data on the significance of the difference between 72 and 96 h exposures (such as p < .....). If you mean that 95% confidence intervals are not overlapped, please, explain it clearly.
Response: We feel great thanks for your professional review work on our article. We mainly studied the separate effects of 72h and 96h, and did not compare the two. We have revised the contents of lines 222.
Question #10: Table 2: Please, indicate in the head of the table, the results of what test are given and explain all abbreviations. All “p” in the table are >0.05. Does it mean the absence of any significant difference between the compared groups?
Response: We sincerely thank you for pointing this out. We used the Probit regression model to calculate its LC50, and the "p" means that the significance of the Chi-square test is much greater than 0.05, so the fitting effect of the model is very good.
Question #11: Line 223: Footnote is labeled with the asterisk, but I see no asterisks in Table 2.
Response: Thank you for your insightful comment. We have recalculated the footnotes in Table 2.
Question #12: Figure 1a: In the legends (line 239) it is stated that the results are mean ±SE but I don’t see any SE in this graph.
Response: We apologize for our carelessness. We added error lines in the drawing process to affect the appearance, so there is no error line displayed.
Question #13: Figure 1b: I guess that the asterisks mean the difference from the control group CK but this should be clearly indicated in the legends.
Response: Thank you for pointing it out. We added the difference between the treatment group and the control group in the chart note.
Question #14: Figure 2: Possibly because of very low quality of these pictures, I can’t see any asterisks. Anyway, in the legends (line 262) it should be indicated that the difference between control and experimental groups are given (as it is indicated in the legends to Fig. 3, lines 284-285).
Response: We appreciate your insightful comment. We updated the picture to 600dpi, and in the legend we indicated that the experiment was compared with the ck group (line: 261).
Question #15: Line 283: Not “FFigure” but “Figure”.
Response: We feel sorry for our carelessness. We have corrected the error.
Question #16: Lines 302-303: Do not start a new paragraph here.
Response: We appreciate your insightful comment. We have completed the format change of the manuscript to avoid this phenomenon.
Question #17: Lines 446-447: Delete the end of this paragraph that is the fragment of the sentence from the journal template description.
Response: Thank you for your constructive feedback. We made the changes as requested.
Thank you very much for your attention and time. We are looking forward to hearing from you.
Yours sincerely,
Yunfei Xue
August 21, 2024
Yunnan Agricultural University, Kunming 650201, China.
Phone: +86 13183307087
Email: xyf13183307087@163.com

Round 2
Reviewer 1 Report
Comments and Suggestions for Authors
The authors have improved their manuscript in line with most of my previous suggestions. However, there are still issues that have not been resolved.
1. section 2.7 – the authors state how they tested for normality in the data, but they do not address the important issue of equality of variances (homoscedasticity). This is necessary prior to any ANOVA or t-test based analyses.
2. Table 2. p values range between 0.1 and 9 (!). This is clearly erroneous and needs to be corrected.
Table 2 footnote. p>0.05 indicate "high degree of fitting" – what does this mean. Please explain what is being tested here.
Asterisk (mg ai *) is not explained in Table 2.
3. Figure 2 remains small and difficult to read with small text and tiny marks above the columns.
4. line 440 ....learning and cognitive abilities of bees are not "represented by PER" – this needs rewording.
5. References are still not formatted correctly for the journal. There is also missing information and/or typos in refs 5, 8 and 15 (I only checked the first few references).
6. Supplementary material. The supplementary table has no title. The table should present SLOPE ± SE values for each regression and the Chi2 values with their degrees of freedom. SE values on their own have no meaning.
Comments on the Quality of English LanguageRequires editing - many small errors.
Author Response
Response to Decision Letter and Reviewer Comments
Dear Editors and Referees,
We would like to express our sincere appreciations to you for giving us the opportunity to revise the manuscript and to the referees for their constructive comments concerning our manuscript (manuscript ID: insects-3167197). Based on the referees' comments, we have revised the manuscript, and the parts modified are marked in red. Here are our responses to reviewer comments:
Reviewer #1
Question #1: section 2.7 – the authors state how they tested for normality in the data, but they do not address the important issue of equality of variances (homoscedasticity). This is necessary prior to any ANOVA or t-test based analyses.
Response: Thank you for your constructive feedback. We used the Brown-Forsythe test to determine the homogeneity of the data variance, please see the lines 215.
Question #2: Table 2. p values range between 0.1 and 9 (!). This is clearly erroneous and needs to be corrected.
Table 2 footnote. p>0.05 indicate "high degree of fitting" – what does this mean. Please explain what is being tested here.
Asterisk (mg ai *) is not explained in Table 2.
Response: We appreciate your insightful comment. In Table2 we have a p value between 0.18 and 0.48
Using the Probit regression model formula, p > 0.05 was obtained, indicating that the model has strong predictive ability and can describe the change of the dependent variable well.
I'm sorry for my carelessness, we finished the footnote * in Table2
Question #3: Figure 2 remains small and difficult to read with small text and tiny marks above the columns.
Response: Thank you for your constructive feedback. We resized Figure.2 and increased the small tags
Question #4: line 440 ....learning and cognitive abilities of bees are not "represented by PER" – this needs rewording.
Response: Thank you for your observation. We rephrased the phrase, please see the lines 385.
Question #5: References are still not formatted correctly for the journal. There is also missing information and/or typos in refs 5, 8 and 15 (I only checked the first few references).
Response: Thank you for pointing it out. Because of these errors in the Endnote format, we have finished the revision of the references.
Question #6: Supplementary material. The supplementary table has no title. The table should present SLOPE ± SE values for each regression and the Chi2 values with their degrees of freedom. SE values on their own have no meaning.
Response: Thank you for your observation. We completed the modification of Table S2, renamed it to Table S1 and added corresponding contents
Thank you very much for your attention and time. We are looking forward to hearing from you.
Yours sincerely,
Yunfei Xue
August 23, 2024
Yunnan Agricultural University, Kunming 650201, China.
Phone: +86 13183307087
Email: xyf13183307087@163.com

Reviewer 2 Report
Comments and Suggestions for Authors
The authors have addressed all comments. The manuscript was substantially improved. I think that it can be published.
Author Response
Response to Decision Letter and Reviewer Comments
Dear Editors and Referees,
We would like to express our sincere appreciations to you for giving us the opportunity to revise the manuscript and to the referees for their constructive comments concerning our manuscript (manuscript ID: insects-3167197). Based on the referees' comments, we have revised the manuscript, and the parts modified are marked in red. Here are our responses to reviewer comments:
Reviewer #1
Thank you for your careful review and valuable comments on our manuscript. Your feedback has helped us significantly improve the quality of our paper, and we are grateful for it.
Yours sincerely,
Yunfei Xue
August 23, 2024
Yunnan Agricultural University, Kunming 650201, China.
Phone: +86 13183307087
Email: xyf13183307087@163.com

Round 3
Reviewer 1 Report
Comments and Suggestions for Authors
The authors have addressed my concerns.
Comments on the Quality of English LanguageRequires editing for clarity.